# Tree Information Modeling: A Data Exchange Platform for Tree Design and Management

Qiguan Shu [1],*, Thomas Rötzer [2], Andreas Detter [3] and Ferdinand Ludwig [1]

1 School of Engineering and Design, Technical University of Munich, Arcisstr. 21, 80333 Munich, Germany
2 School of Life Sciences, Technical University of Munich, Hans-Carl-v.-Carlowitz-Platz 2, 85354 Freising, Germany
3 Brudi & Partner Treeconsult, Berengariastr. 9, 82131 Gauting, Germany
* Correspondence: qiguan.shu@tum.de

**Abstract:** Trees integrated into buildings and dense urban settings have become a trend in recent years worldwide. Without a thoughtful design, conflicts between green and gray infrastructures can take place in two aspects: (1) tree crown compete with living space above ground; (2) built underground environment, the other way round, affect tree's health and security. Although various data about urban trees are collected by different professions for multiple purposes, the communication between them is still limited by unmatched scales and formats. To address this, tree information modeling (TIM) is proposed in this study, aiming at a standardized tree description system in a high level of detail (LoD). It serves as a platform to exchange data and share knowledge about tree growth models. From the perspective of architects and landscape designers, urban trees provide ecosystem services (ESS) not only through their overall biomass, shading, and cooling. They are also related to various branching forms and crown density, forming new layers of urban living space. So, detailed stem, branch and even root geometry is the key to interacting with humans, building structures and other facilities. It is illustrated in this paper how these detailed data are collected to initialize a TIM model with the help of multiple tools, how the topological geometry of stem and branches in TIM is interpreted into an L-system (a common syntax to describe tree geometries), allowing implementation of widely established tree simulations from other professions. In a vision, a TIM-assisted design workflow is framed, where trees are regularly monitored and simulated under boundary conditions to approach target parameters by design proposals.

**Keywords:** tree information modeling; tree engineering; building information modeling; computational design; urban green infrastructure

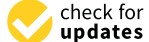



## 1. Introduction

### 1.1. Aim of This Study

#### 1.1.1. History and Trend of Tree Use Integrated in Human Habitats

Integrating trees in nowadays urban spaces as well as buildings is driven by multiple benefits: people's psychological health [1,2], thermal comfort in the context of the urban heat island (UHI) effect [3,4] and sustainability [5]. In architectural history, trees were already used at an early stage of settlement to provide shelter against flood and beasts. Later historical cases saw trees utilized more multifunctionally: espalier trees are trained in geometrical forms for acquiring structural stability and increasing fruit yield in the city [6] (i.e., an espalier tree in England shown in Figure 1c); through pollarding and coppicing, trees were manipulated to produce firewood and building materials [7] (i.e., pollarded trees were bent into umbrella shapes in the center of Labouheyre, France to shade the public square shown in Figure 1d); Devon hedges were built by injuring and laying down trees on earth banks to protect cattle or crops [8,9]; vite maritata systems used trees as supports for vines, whilst providing wind- and sun-protection for field crops and an ecosystem for

diverse dependent species [10]; Hausschutzhecken are trees weaved into stable structures to protect buildings and gardens from wind [11]. Street trees nowadays, however, are mainly kept in freely growing forms to reduce maintenance costs and failure risks [12]. Owing to this situation, studies regarding their ecosystem services (ESS) focus on certain aspects such as cooling, shading, carbon storage and reduction in rainfall-runoff [13,14]. But beyond these, from the perspective of architects and landscape designers, trees can provide further values in ESS with specially trained forms.

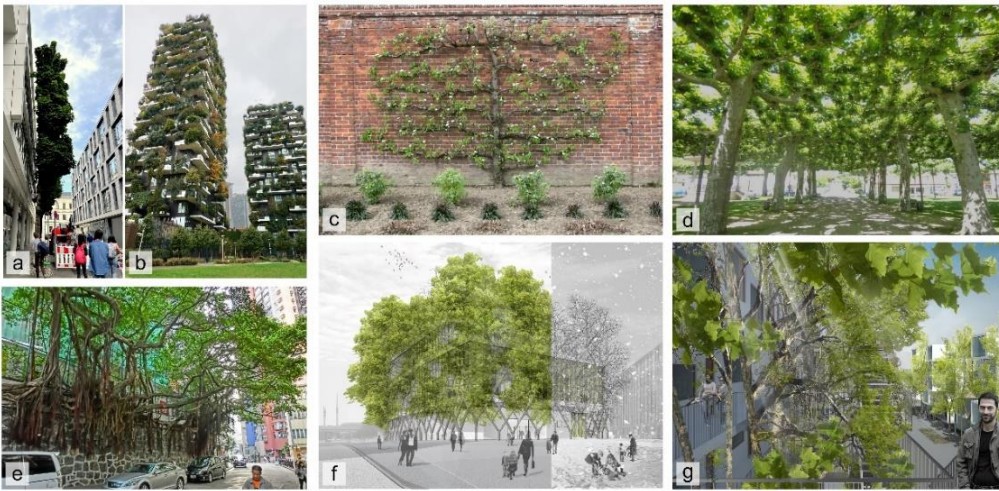

**Figure 1.** Challenges and potentials of trees in dense urban areas. (Image (**b–e**) are retrieved from the internet under Creative Commons licenses. Image (**a,f,g**) are own images).

Interest in multifunctional trees in public spaces and in the building context has been growing in the architecture and landscape architecture industry: in Figure 1a, a tree is planted close to a building façade in Munich, growing only one side of its canopy; in Figure 1e, ficus trees that take vertical walls as their foundation are preserved in Hong Kong. Besides, more proposals are framing trees as their core concept in recent international design competitions; for example, Madrid's RENAZCA development by MVRDV [15] illustrates a public space surrounded by luxuriant "floating" plants grown on metal scaffolding; Street Tree Pods by Matthew Chamberlain [16] offers single apartment on street trees to alleviate London's housing crisis. Buildings with trees integrated into the envelope appear recently in practice, like Bosco Verticale in Milan, designed by Boeri Studio (see Figure 1b) [17] and Kö-Bogen II in Düsseldorf designed by Ingenhoven Architects [18]. Living Architecture, especially Baubotanik [19] (i.e., the house of future proposed for a museum in Berlin by Ludwig Schönle, shown in Figure 1f), go one step further, exploring trees as load-bearing structure. In Figure 1g, the tree façade forms a vertical open space. In this way, the ESS of urban trees lies not only in their general biomass and canopy volume for increasing biodiversity and thermal comfort but also in configurations of roots and branches as a sustainable material to enclose, support and co-create living spaces.

### 1.1.2. Conflicts between Gray and Green Infrastructure

This trend to enhance trees' multifunctional use in the building industry is confronted now with conflicts from two sides: on the one hand, trees physically can reach a height of at least 10–25 m [20–22] in cities. This height range is also occupied by multi-layer traffic systems and the pedestrian bridges like those in Hongkong [23] as well as public spaces like the High Line in New York City [24]. Within 25 m height are also common residential buildings of 7 floors. Therefore, free-growing tree canopies are competing with these building structures in space above the ground. Too densely aligned canopy can reduce street ventilation, causing traffic-related pollutant concentrations [25,26]. On the other hand, densely built underground environments led to a high removal rate for urban trees. Although some tree species can live up to 200 years in principle, most of them

would not be retained longer than 40–60 years in cities [27] due to damages caused by humans [28] or low mechanical performance. Consequently, the average lifespans of urban trees are shorter than the average operating stage of residential housing (61 and 120 years in the US [29] and Denmark [30], respectively). If trees involved in a building structure should not be "temporary" installations but accompany the whole operational period of the building, thoughtful tree design and management are in urgent demand. Healthy and secure growth of individual tree branches and roots must be wisely integrated with urban gray infrastructures (e.g., building façade, foundations, underground pipelines, and even subways).

### 1.1.3. A Novel Workflow for Tree Design and Management

In this scenario, project planning and maintenance will play a key role in its success. As tree growth is complex and dynamic, the chance to precisely predict and control this process for every branch is small. If pruning all unwanted branches away constantly, this does not effectively reach the purposed functional use provided by specific configurations of branches. Therefore, tree design and management must take place through the whole life cycle in those tree–closely–integrated structures. As illustrated in Figure 2, a typical contemporary architecture design is a set of definitive solutions based on boundary conditions and clients' requirements. If unforeseen circumstances occur during the construction, designers usually seek minor adjustments in response to the problem. The form of the original building design, at least, will be maximally preserved. This common workflow is seen in most of nowadays building projects. But to design and build with living trees, projects by Ludwig and Schwertfreger [19] followed an iterative design approach. It means designers repetitively check tree growth (every 1–3 years) and accordingly make adaptions to the design proposal.

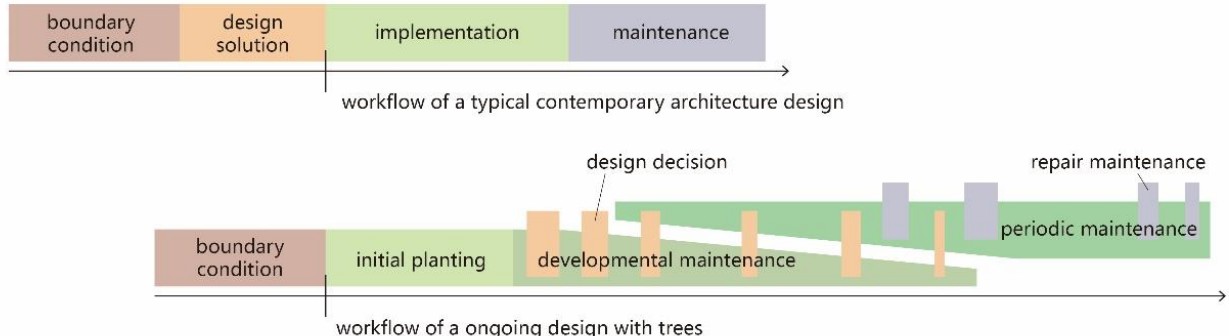

**Figure 2.** Proposed workflow for a dynamic design to deal with tree growth and death. Unlike a contemporary architecture design (shown on the **top**), the design decision (see dark orange rectangles) in the new workflow will no longer be made only once but multiple times through the tree's life cycle (shown on the **bottom**).

However, traditional tree management is either empirical-based labor-intensive work or standardized periodic pruning [31]. Under current conditions, tree management has a high demand for labor resources. It is not feasible to implement such workflow widely only with human forces, especially when more buildings embedded with trees are being built. To solve this problem in the future without losing the goal of achieving both high ESS values of urban trees as well as their long-term healthy living, efforts must be made in an automated tree management system. With the rapid development of machine vision and automatic robots, urban trees could be taken care of by (swarms of) robots as a low-cost working force in the future [32]. Meanwhile, the public might also be interested in participating in managing trees (including harvesting fruits) in the community as their recreational activity during their free time. Such participation in managing urban green infrastructure by either robots or non-professional personnel requires an appropriate knowledge base about tree management. This knowledge base must incorporate at least the following topics: (1) the

impact of the urban environment on tree growth in various LoD; (2) the impact of human intervention (i.e., pruning and bending branches) to tree behaviors; (3) evaluation index for ESS of urban trees in different professions and aspects; (4) efficient and low-cost solution for tree measuring, modeling, and manipulation; (5) solution for measuring, modeling and maintaining buildings and other gray infrastructures in a more dynamic context; (6) design strategy and practice how trees best co-live with buildings and other urban facilities.

### 1.1.4. This Solution Requires Interdisciplinary Cooperations

These topics are already partly covered or being studied in multiple fields like urban forestry and arboriculture. But different professions, although all have their general interests in trees, aim at completely different goals (see details in Section 1.2). For example, forestry scientists analyze trees' role in an ecosystem; arboriculturists ensure trees' safety and health; biologists simulate the physiological process of tree growth; computer graphic experts contribute to tree geometries and visualizations. For such purposes, they independently collect relevant data about trees to build their own models in various scales, LoD and functions. However, cross-disciplinary communication remains limited owing to unmatched scales and data formats. Cross-platform media, which allows data exchange and model sharing between tree-related professions, is of great importance.

The building industry experienced a similar problem when data from architecture designers, civil engineers, constructors, and equipment suppliers did not communicate. Even on the same industrial chain, data were repeatedly collected and semi-manually transferred from one model to another for different purposes until Building Information Modeling (BIM) was developed. Standards were set for integrating digital models and properties of a building system, such as water, electricity, and gas infrastructure/installations [33]. Such digital copies of building systems also provided an interface for adding dynamic operational data, e.g., via smart meters. These data, referring to the physical entities in the real world, are now becoming the base for the Internet of Things (IoT) [34,35]. In an urban scope, this idea of establishing a data sharing and exchange platform leads to digital twins of cities whose plan, design, construction, and operation can be guided and driven by data to improve working efficiency and quality while bringing down energy consumption and waste. City Information modeling (CIM) is being discussed and developed in several cities around the world [36], for example, Digital Twin Munich [37] and Chinese Xiongan new area [38]. Landscape Information Modelling (LIM) was also proposed for efficiently managing projects in the landscape architecture industry [39]. But until now, a standard media for sharing data and models of urban trees is absent. This standard must reach a certain LoD to enable participation from more professions. If in a low LoD, generic indexes proposed in LIM and CIM, such as tree height, diameter at breast height (DBH), and canopy diameter for describing one tree, cannot guide tree manipulation in arboriculture and cannot integrate the physiological tree growth model from biological studies.

### 1.1.5. A Starting Point for the Proposed Workflow

This study, therefore, put forward the concept of tree information modeling (TIM), aiming at a data exchange and model sharing platform for gathering cross-disciplinary knowledge about urban trees. For architects and landscape designers, this is a starting point to bring trees and building structures in harmony. To achieve this aim, we investigated interdisciplinary fields (see Section 1.2) to propose a unified tree description system consisting of information tags and geometrical representations of trees. This modeling framework meets the scope of future requirements for tree design and maintenance introduced above. Several example methods for data acquisition (Section 2.2.1) and data interpretation (Section 2.2.2) are described in further detail. With TIM, it is expected to establish a survey-simulation-manipulation workflow for managing future urban tree systems (Section 3.3).

*1.2. Related Studies in Interdisciplinary Fields*

1.2.1. Tree Management and Risk Assessment in Arboriculture

In current arboricultural practice, urban trees in public are often managed in this manner [40]: for young trees, branches at the lower side of the canopy will be trimmed off to allow for traffic to pass under, which is called lifting or crown raising; before the trees reach a certain height, part of dense branches can be removed to avoid collision and to reduce competition between leading branches for space and light, which is known as crown thinning; any branches growing close to power lines, traffic lanes and private spaces, are often removed or reduced in length, which is called crown reduction. Current best practice includes many more arboricultural measures and specifications for their application (e.g., ATTC [41], Lilly, Gilman [42]).

For all these approaches, pruning stands at the heart of arboriculture [43]. Pruning not only alters the short-term appearance of trees but impacts phytohormone (i.e., auxin-cytokinin) distribution inside them (i.e., apical dominance [44]). Therefore, which branches to be pruned demands craftsmanship. For instance, when a tree fork consists of two sub-branches of comparable size and angle, one of them may be pruned to slow down its growth. After 5–10 years, as a result, the reduced branch on this fork will be clearly subordinated. Otherwise, a fork union with stems of similar diameter may evolve, which forms a weakness in the tree's mechanical stability [45].

For arborists, safety is always a priority in conducting tree management (compared to health and aesthetics listed by Bedker [40]). Root failure and decay inside the trunk are two major causes of tree risks [46]. Due to their high complexity, both risks cannot be precisely analyzed by computational simulations yet. Respectively, to rule out root and trunk failure, practical pulling tests are developed to measure the tree's tilt and deformation using an inclinometer and elastometer under a given force on the main stem [47,48]. This method bypasses the technical obstacle in detecting actual root geometry and decay locations as well as understanding the force transmission from the roots to the soil or within the stem's geometry. To spot weak spots along trunks, an empirical visual inspection could solve the vast majority of cases. The rests require assistance from advanced methods of tree assessment, for example, tomography based on either the time of flight of sound waves or the electrical resistivity of wood. These tomograms display the distribution of resistivity across the stem cross-section, which can indicate featured patterns of decay taking place [49].

Studies have been carried out in recent years about how these data can be collected on a large scale in real-time for monitoring tree tilt angle and sway under natural wind loading [50]. Integrated with GIS, these data can be utilized more fruitful than studying tree risks. The sensors installed on a huge number of trees across the city become a network of mini weather stations, which can be used to precisely model near-earth wind speed and wind-load effect of trees [51]. Data collected and studied in arboriculture do not rely on the detailed geometric representation of tree branches and roots at the current stage. But other professions have different situations.

1.2.2. Urban Tree Models and Databases from Forestry Science

The rise of urbanization since the 1800s has driven various professions to investigate the hybrid system of man-made environment and nature: In architecture, Frederick Law Olmsted and Calvert Vaux [52] for the first time used the term "landscape architecture" in 1863 for a new profession, which extended traditional architecture studies to open space systems. Natural ecosystems, in this way, gained a place in urban-scaled planning. Forestry science developed its scope in the opposite direction. It started by managing natural resources in the natural environment but then faced the challenges of re-evaluating such resources in an urban context for making public policies. To merge this gap, "urban forestry" was first invented in North America in 1965 to integrate a broader group of experts (e.g., psychologists and sociologists) in forestry education [53,54]. Urban forestry is now commonly recognized as the sub-discipline for managing trees, other vegetation

and water resources in urban ecosystems for benefits in multiple aspects like sociology, economy, and aesthetics [55]. The integration of all these aspects holds the overall goal: amenity and the promotion of human well-being [56,57]. This is how both urban planning and forestry science meet in this field.

In forestry studies, foliar and woody biomass are the two most important indicators to evaluate a tree's long-term contribution to its ecosystem. Accordingly, several key parameters are widely documented to describe tree stands: trunk diameter at breast height (DBH) and tree height are key factors for estimating the woody biomass; leaf surface area [58], height and diameter of the crown are used for calculating the foliar biomass [59]. Empirical equations are summarized from long-term forestry investigations to predict biomass increment of different species in specific climate zones at different ages [60]. Such equations need to be adapted to urban contexts owing to higher air temperatures and less precipitation in high-dense areas [61]. Therefore, recent development adds further data to describe the surroundings, for example, tree distance from buildings and whether tree crown conflict with overhead wires [62].

The most popularly used urban tree database, including numerical models, is now i-tree, developed by the United States Department of Agriculture (USDA). It was originally the Urban Forest Effects model (UFORE) in the 1990s before the concept of "ecosystem services" was brought out [63]. Ecosystem services (ESS) are defined as the functional components of urban greening that are directly enjoyed, consumed, or used to produce specific, measurable human benefits [14]. Under the demand of quantifying ESS in its subcategories (provision, regulation, support and cultural services), urban tree growth models today include a variety of empirical (e.g., i-tree) and process-based (e.g., CityTree) equations that encompass trees on urban and rural lands for estimating their performance such as cooling, pollution mitigation, stormwater run-off reduction, carbon sequestration and storage [64,65]. Besides this, databases supporting tree selection are also developed [66].

In brief, these models, as well as the databases developed in the field of urban forestry, are global descriptions of the trees, where relatively reliable top-down simulations (see Section 1.2.3) are built but not knowing detailed physiological processes among tree organs and branches.

### 1.2.3. Functional Structural Plant Models in Varies Scales

Plants are a typical complex system [67]: this system has both biotic- (e.g., leaf, stoma, and cell) and abiotic environmental (e.g., light, water, and nutrients) elements that interact [68,69]; these interactions can be physical, chemical and in other forms, but are often interlinked, resulting in partly deterministic partly stochastic performances [70]; when observing an overall outcome of the system (e.g., growing direction of the shoot), however, certain patterns would be recognized as a result of emergent behaviors [71]. To describe the macroscopic outcome of trees like the total biomass increment, Top-Down models (i.e., empirical equations) are feasible to predict the general growth tendency; but for understanding emergent behaviors (i.e., branches competing for light), Bottom-Up models work with more similar principles as natural phenomena and processes.

While forestry scientists, by studying wood production on a large scale, developed top-down models (already described in Section 1.2.2), botanists, by studying plant physiology on a micro scale, developed bottom-up plant growth simulations. Since the 1990s, the term 'functional structural plant model' (FSPM) has been used to describe such bottom-up models [72], which contain descriptions of metabolic (physiological) processes that are combined in the presentation of the 3D structure of the tree [73]. After more than two decades of development, FSPMs are gradually studied and applied in multiple fields (i.e., biology, animation, forestry and agronomy) in various scales ranging from meristems to plant communities [74]. While studies of metabolic processes are mainly concentrated on several key aspects (like water uptake, transport, photosynthesis, etc.), plant structure is represented variously in different scales.

On a population-to-plant scale, plant structures are summarized as global presentations [75]. An overall geometric shell (mostly a sphere, ellipse, and cylinder) is used to represent the size and volume of the tree canopy and trunk. This is already capable of calculating several fundamental interactions of a tree and its environment, such as a rough light reception [76] and wind force on the canopy [77]. To understand basic carbon distribution [78] or water transport [79] between pools of leaves, roots, fruits and stems, conceptual compartments are built to set individual equations for different pools. But these plant structures [78,79] remain at a global level. In smaller scales, such as the scale of one plant or the scale of plant organs, the applied plant architecture is summarized as a modular representation [75]. The module can be either a spatial cell, geometric cell, or topological cell. The spatial cell is voxel to illustrate the spatial occupation of the objects, i.e., leaf area density [80]; the geometric cell uses a common set of parameters to describe shapes of similar elements, i.e., leaf length, radius and size [81]; topological cell indicates exact connections between plant organs [82]. Topological models of a tree, due to their unique ability to bridge plant organs and the individual plant [83], are framed in most of FSPMs on a plant-organ scale (see also Section 1.2.4).

### 1.2.4. Quantitative Structure Models (QSMs) of Trees

Topology describes the properties of a geometric object that are preserved under continuous deformations, such as stretching and twisting, but not tearing or gluing [84]. For studying an object with complex geometry (compared to a simple geometric solid such as a cube or sphere), a "thin" version of the shape is commonly used for representing its geometrical and topological properties, i.e., its connectivity, length, and direction, in an easier form. Such an abstraction of the shape that is equidistant to its boundaries is called a topological skeleton [85]. Its mathematical definition varies from distance function [86] and medial axis [87] to morphological operators [88]. Despite these different types, the skeleton, together with the distance of its points to the shape boundary, contains all the necessary information to reconstruct the shape.

The essential format for skeleton data consists of vertices (also called nodes or points) which are connected by edges (also called links or lines). In this perspective, a topological skeleton is also a graph (a mathematical structure [89], which can model pairwise relations between objects [90]. For modeling trees specifically, vertices represent buds, apexes, and nodes for locating other plant organs (i.e., fruits, flowers, and leaves); edges represent internodes, the trunk and branches; a combination of these vertices and edges can form growth module like apical meristem to perform a certain metabolic process such as blossom or elongation [91]. This tree graph has three possible computational data types [75]: a chained list of records that use a single pointer at each child node pointing towards its parent node [92]; an incidence matrix with each vertex in a column, each edge in a row and a number to indicate their relationships [93]; strings of characters that use specific marks to encode graph architecture [82]. Due to its convenience in reading, rewriting, retrieving, and calculating, the plain text string has become the most popular computational data type for FSPM studies. It was a powerful medium to convey topological information when the computational power was limited compared to today [94].

The way strings of characters encode plant structure is called L-system [95]. Based on L-system as the general approach, multiple FSPM platforms have been developed: L-studio coded in C++ provides a library of programs for simulating environmental processes that affect plant development [96]; GroIMP based on the relational growth grammar coded in java enables parallel modification of the geometries while performing the rewriting rules [97]; OpenAlea achieves a graphical programming environment in python offering FSPMs to a larger range of audiences [98]; most recently, L-Py further improved the flexibility of building FSPMs in python and kept compliance with other platforms [99]. Varies plant structures are built, such as Kiwi fruit [100], peach [101] and apple [102,103], in relation to multi-aspects of the metabolic process like photosynthesis (see kiwi), water stress (see peach), pruning (see apple), gravity and light competition [104]. Despite substantial

results of these studies using L-system, strings for describing the topology of trees have certain limitations: firstly, interpreting 3D structures of real trees with massive nodes through linear string has low fault-tolerance; secondly, L-system cannot describe re-joint branching networks such as inosculated tree structures [105].

In the past ten years, the popular use of Terrestrial LiDAR Scanning (TLS), structure from motion (SfM) [106] and rapid growth in computing power for 3D graphics enabled detailed documentation of objects with point cloud data. In the field of remote sensing and computer graphics, these technologies were soon applied to tree surveys. 3D geometric primitives of trees can be abstracted from discrete points to represent the structure and topology of their trunk and branches [107]. Multiple approaches were developed for this purpose: Raumonen [108] developed their own method using "cover sets" to reconstruct tree topology; similar to the cover-set idea, PypeTree [109] rebuilt trunk and branches by their "segment" based on skeleton curves in python and then used semi-supervised adjustment to correct the errors; SimpleTree [110] built cylindrical tree models in C++ by voxel-grid and Euclidean clustering, it also developed crown calculation tool to estimate canopy volume; cylinder fitting was proved robust in shape fitting for tree trunk and branches [111]; AdTree [112] was another skeleton related approach fitting cylinders to point cloud model of a single tree. These solutions to generate QSMs enable physical trees to be converted digitally.

Alternatives for geometric tree branch primitives are seen in the gaming and animation industry. Animation rigging of characters [113] is also adapted to plant models by motion capture [114]. Trunk and branch segments can be defined as rigid bodies connected with constraints. In this way, force (i.e., gravity and wind) and collision can be calculated by physical engines [115].

1.2.5. Digital Tools and Databases of Trees Used in the Building Industry

Following the trend of digitalization driven by rapid IT development, the building industry is also transforming into digital tracks at all its stages, including planning, design, execution, and management [116]. Formulated data and methods also vary for different working scales and purposes (see Table 1).

**Table 1.** Comparison of digital tools and tree representations in different scales.

| Scales | Urban | District | Single Build Project |
|---|---|---|---|
| **topics and purposes** | Land use and planning | Thermal comfort, ecosystem services | Structural performance, operation, and maintenance, building economy |
| **tools and databases** | GIS, CityGML | Environmental design-decision-support platform | Building Information Modeling |
| **Suitable models for trees** | Population model, Raster image | Global representation, Spatial decomposition (voxel cells) | Topological skeleton Cylindrical pipes |

In the scale of urban areas, Geographic Information System (GIS) offers a platform to overlay both raster images by satellite remote sensing and vector data by field mapping [117]. Vegetation in this scale is represented by the leaf area index (LAI). It indicates leaf area per unit ground surface area, which is estimated by normalized difference vegetation index NDVI [118,119], measured with red and near-infrared regions of the electromagnetic spectrum. This index is used to document and analyze the change in plant populations on a large scale. Recently, LAI is also used to guide urban development in terms of green space and urban forestry [120]. CityGML is an advanced all-in-one database in open standardized XML. It enables potentially describing all city facilities (3D objects) in various LoDs [121]. Gobeawan [122] very briefly proposed four levels of tree representations in CityGML, namely a plane circle, one single cylinder, a convex hull and detailed leaves

and branches. Among these levels, tree models in LoD 1–3 are already seen in multiple CityGML databases, while LoD 4 remains unclearly defined.

On the scale of district and community, design-decision-support platforms [123,124] are built to simulate interactions between buildings, plants, environmental conditions, and human activities [125]. Kirnbauer [126] integrated multiple databases into a decision support system for urban tree planting. For these purposes, trees are mainly represented by their canopy volume and position using either a simple geometry [127] or voxels [26]. In this way, environmental engines like ENVI-met [128] and Grasshopper plug-in PANDO [129] can simulate the tree's shading and air flow affected by the tree canopy. These assist decisions in street section or plant arrangements for outdoor comfort.

In the scale of a building and its construction, building information modeling (BIM) has been developed to integrate all necessary information concerning building facilities through their lifecycle [130]. Such data can be as detailed as materials and their manufacturers. Some attempts are made to utilize BIM on the district scale, where global tree representations (similar to Vos et al. [26]) appear alongside a BIM model [131]. However, trees are not yet regarded as core components for buildings that can be integrated into BIM and utilized for living architecture design and engineering. No unified description is given about the data type, utilization, and purpose of tree models in this scale for architecture design. Therefore, this paper provides one standard and solution to fill this gap (see Section 2.1).

## 2. Tree Information Modeling

### 2.1. Definition

Similar to the definition of BIM [132], Tree Information Modeling (TIM) is conceived as a digital representation of the physical and functional characteristics of a tree. We define TIM as a data exchange and knowledge-sharing platform about trees, aiming at a solid basis for decision-making in their planning and life-long management. To avoid disaccord understandings that occurred on BIM during its long development [133], in this paper, TIM does not limit to any specific tool or software to convey and calculate the tree data. It is a framework following the same tree description system (TDS) to create the digital twins of trees in real life. A unified updating version of TDS enables the maximum compatibility of all TIM users. The first version of the TDS is stated below (see also Figure 3): A digital tree consists of basic information tags and a geometric representation; basic information tags should include at least tree species, tree age (or years after the first planting), location by longitude and latitude, date of documentation (not necessary for a virtual tree at its planning phase). Information tags must also support additional attributes such as tree images, transplanting history and results of pulling tests in risk assessment. Geometric representations have three compartments: branch (including trunk), leaf canopy, and root; trunk and branch (incl. aerial roots visible above the ground) are represented by the topological skeleton and cylindrical pipelines; leaf canopy is represented by voxel noted with leaf area density; root underground is represented by iso-density layers. Each geometric element can be attached with additional attributes if they are measured, such as decay, sap flow rate, the concentration of phytohormones and electrical resistivity.

It needs to be clarified that due to current limitations in underground detection, even the proposed rough iso-density layers for representing roots cannot be correctly mapped in practice.

It is also aware that even if such a root model is acquired, this may not be sufficient to serve all demands in analyzing tree roots (i.e., root failure described in Section 1.2.1). We propose this root representation in the current version of TDS as a balance between what needs to be studied and which data could be gathered. Depending on the technology development, the root model could be updated to root density voxels in later TDS versions or even cylindrical pipelines (the same as branch representations) in the future.

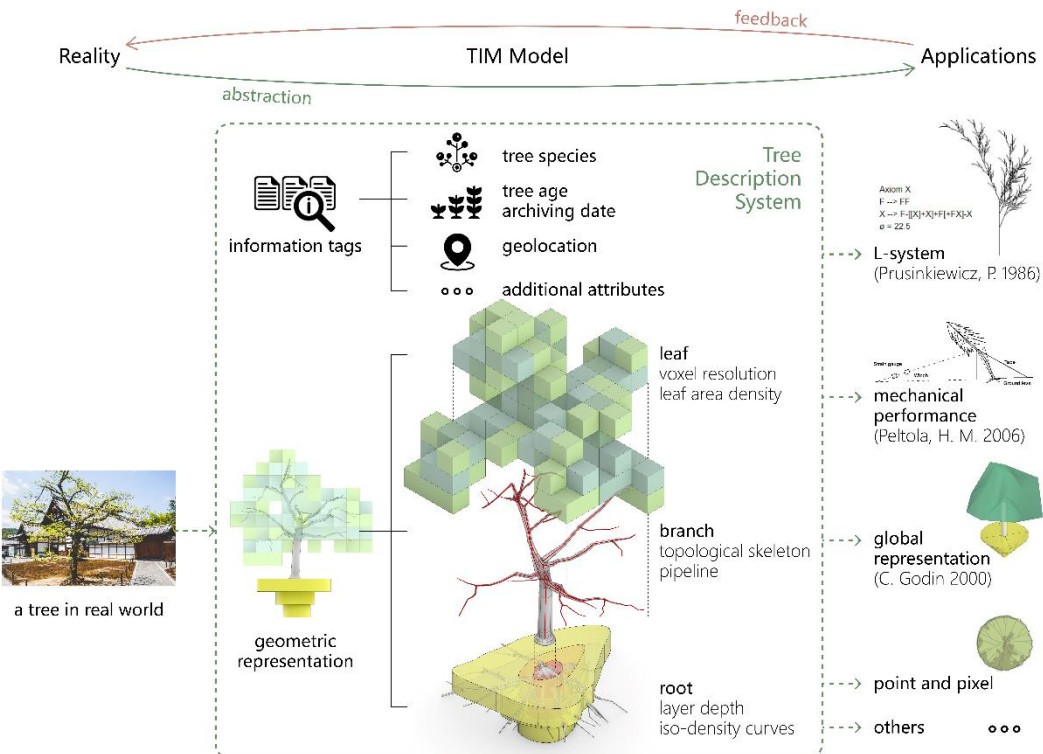

**Figure 3.** Role of TIM and its data structure. TIM integrates tree information required in multiple professions (see e.g., [48,75,95]). It functions as a media between trees in reality and their digital twins for specific applications.

As for the relationship between BIM and TIM in the building industry, both models function as media between physical objects in the real world and digital applications for planning, evaluating and maintenance. An overview of their geometric representations and key parameters for applications were listed in Table 2. TIM can be applied independently from BIM to digitalize living trees. Data formats in TIM are set with already supported data in BIM software: voxel for leaves, the pipeline for the trunk and branches and closed splines for roots (as shown in Figure 3). When buildings in the future take trees as components and even require their interactions with other building facilities, TIM can be integrated into BIM for more comprehensive uses. This relationship also applies to LIM and CIM on a larger scale, where TIM could benefit the urban green system planning and management as well as evaluate and forecast trees' impact on the urban environment.

Most importantly, although no study about trees yet is built on this novel definition of TIM, TIM models, once initiated or even partly initiated (i.e., missing root data), can be directly applied to related professions introduced in 1.2 because TIM has stored the geometry of trees in a high LoD. Other professions working on more abstracted geometric datasets can interpret TIM into their corresponding forms, such as L-system (see Section 2.2.2), rigid body, a sphere canopy or a pixel. In this way, existing studies, and methods in the fields of forestry science, FSPM and building environment (see Section 1.2) can all be applied to TIM. In this perspective, TIM works as a platform for merging these studies.

**Table 2.** Comparison between common tree models used in multiple fields and TIM.

| | Common Geometric Representation (for Structural Model) | | | Common Physiological Parameters (for Functional Models) | Common Environmental Factors |
|---|---|---|---|---|---|
| | **Leaf** | **Branch (Incl. Stem)** | **Root** | | |
| **Forestry science** | Crown as an elliptical sphere by its height and diameter | Trunk as a cylinder by DBH and crown height | Not involved | Leaf surface area; sap flux; | Climate; temperature; population density |
| **FSPM** | Individual leaf as a rectangle by its length, width, and position | L-system with turtle interpretation | L-system with turtle interpretation | Water transit; carbon assimilation and allocation | Gravity; light rays; |
| **Mechanical calculation** | Windward area, leaf density and drag | Trunk as a unilaterally fixed, tapered cantilever beam | A joint with viscoelastic properties | Not involved | Wind velocity, temperature, moisture content |
| **Land resource management** | Leaf area index in pixels | Not involved | Not involved | Not involved | Near-infrared spectroscopy; red spectrum; |
| **BIM** | Crown as an elliptical sphere | Trunk as a cylinder | Not involved | Not involved | Not involved |
| **TIM** | Voxel by leaf area density | Topological skeleton and pipelines | Layer by iso-density curves and depth | Water transit; to be developed | To be developed |

*2.2. Methods for Data Acquisition and Processing*

2.2.1. From Reality to TIM—Data Acquisition through Multiple Tools

As described in Section 2.1, a complete TIM model consists of information tags and geometric representation. In the planning phase, virtual data could be directly generated through designing and simulation programs and then fed into TIM. In this case, it is recommended to initiate a field named "virtual" in the information tag and set its value to true. In the maintenance phase, creating a digital twin of one physical tree must apply different methods in gathering required data: for collecting the information tags, geolocation (longitude and latitude of the tree) can be recorded by portable GPS devices at the stem base; tree species, age and archiving data require manual entries before an automatic tree identification program (i.e., possibly driven by deep learning [134]) is developed; a few photos shot in different distances and angles could also be attached as additional attributes for training the automatic tree identification and for manual cognition. For creating the geometric representation of a tree, the branch (incl. trunk or aerial root) and leaf are visible compartments above the ground, while the root is invisible beneath. So, the surveying methods are different.

Documenting the topological geometry of branches consists of 3 steps (see Figure 4). (1) By LiDAR or photogrammetry scanning [135], a point cloud model of the visible compartment (trunk, branch, aerial root and leaf) can be created. At this step, improvements can be made regarding computing time and tolerance to point clouds in low quality (see Section 3.1). (2) To abstract the topological skeleton of branches out of the point cloud, Cornea [136] compared multiple automatic skeletonization methods; L1-medial skeleton [137] is efficient on point cloud that is not over complex containing too large an amount of points; [138] developed an approach to restore a speculative skeleton without segmenting point clouds into branches and leaves; Wu et al. [139] then achieved an accurate median-axis skeleton abstraction based on the foliage–woody separation by convolutional neural networks [140]; Liu et al. [141] developed a neural network to reconstruct tree geometry out of a point cloud robust to noise, outliers and incompleteness; besides, voxel thinning is able to preserve the precise topological structure of tree branches while estimating approximate

diameters of branches during the thinning process [142]. (3) After skeletonization, pipelines can be generated by cylinder fitting or calculating the average distance from the trunk surface to the skeleton on perpendicular planes.

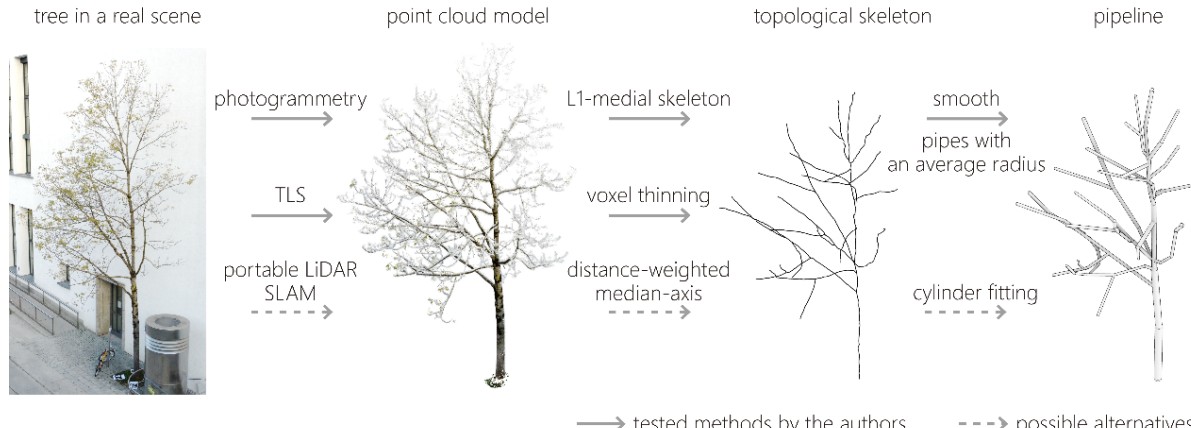

**Figure 4.** Process to collect topological geometry data of the tree trunk and branches.

Figure 4 illustrates these steps with a tree standing close to a building façade. The arrows with solid lines show several technical roots that are tested by the authors. Arrows with dotted line show alternatives to the same functions. The displayed tree grows an asymmetric canopy against the wall. In this case, sphere geometry as a global representation of a tree canopy is not a precise way to describe it. TIM has the advantage of documenting its main trunk and detailed branches with QSMs.

Voxel-based descriptions of tree canopies were first developed to represent only the volume [143]. To derive LAD at the voxel scale, the Monte-Carlo simulation is a classic approach [144]. Béland, Widlowski [145] proposed the VoxLAD model instead of ray tracing algorithms, enabling the estimation from discrete returning data from any type of TLS; Wu, Phinn [146] used this method on multiple species of fruit trees; Hosoi and Omasa [147] developed voxel-based canopy profiling method to estimate LAD in voxels. The precision according to the voxel size is assessed by Li and Dai [148].

TLS is also applied in scanning tree roots if roots are dug out from the ground [149]. But to detect roots underground, ground-penetrating radar is used. It transmits and receives electromagnetic waves. The returning signals indicate boundaries of overlaying objects [150]. The precision and maximum depth depend on the wave frequency and soil type. Inhomogeneous soil, commonly seen in urban areas, usually produces poor results. New methods like multi-electrode resistivity imaging used for detecting decay inside trunks (see Section 1.2.1) [151] can also show a rough distribution of roots underground, but they are not yet applicable in practice. These data can be processed into 3D root layers in CAD software (i.e., see Gärtner et al. [149] Section 3.2).

Establishing a complete TIM model requires a combination of all methods above. A tree survey is recommended to be conducted in different seasons. For deciduous trees, for example, their LAD can only be documented in summer, while their trunk and branch geometry can only be documented in winter when there is no leaf. Due to such a high standard for completing a TIM model, an incomplete TIM dataset will exist for future applications. Therefore, their access to certain functions should be checked if the required data is missing.

2.2.2. From TIM to Established Applications—Interpreting Pipelines into L-System as an Example

Once trees are documented in TIM, all professions listed in Section 1.2 can extract part of the data from TIM to build their own established model for analysis and simulation. For forestry scientists, for instance, DBH is the diameter of the pipeline at 1.3 m height above

ground; tree height is the *z*-axis coordinate at the upper side of the top canopy voxel; total biomass estimation is the mass sum of leaves, branches, and roots using their volume and average density. Such calculations are similar to measuring real trees, therefore, will not be further explained in this paper. For FSPM studies (see Section 1.2.3), L-system is most used in plant growth simulations. So, it is important to illustrate here how to interpret data, especially branch geometry in TIM, to a tree model written in an L-system.

L-system is a string rewriting mechanism. It recursively replaces certain parts of the strings according to given rules. In this way, it produces patterns with self-similarity, thus being widely used for modeling plants [82]. The method to draw the geometry based on the commands in strings is called turtle interpretation [95]. Typical commands direct a virtual "turtle" moving forward or turning its heading. The trail of the turtle is the geometry to be drawn. However, the original version of such a symbolic L-system has limitations in (1) setting an individual rotating angle and distance for each move, (2) implementing physiological functions for plant growth and (3) operating on the drawn geometry. Therefore, some improvements were developed later (see Section 1.2.4). To illustrate the interpretation from pipelines into one of the L-system models, this study takes language XL as the target format. Language XL is an implementation of relational growth grammars (RGG) [97]. It enables parallel plant description rewriting and geometry generation [152]. Pipelines in TIM can also be interpreted into other variants of the L-system following the same approach but making adaptions to format writing.

Interpreting pipelines into language XL has three steps: (1) translating each pipeline; (2) combining branches in the order of topology; (3) adding defined tree organs to the model.

In the first step, each pipeline in TIM is defined with its geometry and spatial location. The pipeline's geometry consists of its length *l* and diameter *d*, which are the same parameters to draw cylinders in XL language using the command $F(l, d)$. The spatial location of a pipeline in TIM is marked with its two ends (i.e., $A(x, y, z)$ and $B(x\prime, y\prime, z\prime)$). In XL language, location of an object depends on turtle's state including its position and heading. Turtle's state initiates at the origin point with the default heading shown in Figure 5a. This state will be updated in each step by moving and rotating. Rotation is described by Euler angles along the turtle's local X, Y and Z axis (see also Figure 5a). As cylinder is central symmetric, two degrees of rotation could reach any demanded orientations in a 3D space. In this section, we will use only the turtle's Z and Y axis to perform Euler rotation. As shown in Figure 5b, the command $RH(\alpha)$ and $RU(\beta)$ in XL language rotates the turtle's heading $\alpha$ degrees along the Z axis and $\beta$ degrees along the Y axis, respectively, in the illustrated direction. Their values can be calculated with equation 1 and 2. A negative number indicates the rotation in a reversed direction. As the rotating sequence affects the results, Z-Y is ruled for all rotating sequences. Starting from the default turtle heading, the rotating angles are calculated with equations 1 and 2. After moving forward along the pipeline, the turtle should make a reversed Y-Z rotation to return to the default turtle heading. This step is crucial for easier connections between individual pipelines because the turtle's headings are identical for all elements. In all, the outcome command for one single pipeline is formed as follows: $RH(\alpha) \ RU(\beta)F(l, d) \ RU(-\beta)RH(-\alpha)$ (see Figure 5c). Such a set of commands for drawing one single pipeline is noted with $p_n$, where *n* is the number of the pipeline. When a movement of the turtle is required without drawing a cylinder, the command $F(l, d)$ can be replaced with $M(l)$, where *l* is still the distance of moving. In this case, the command series is written as $RH(\alpha) \ RU(\beta)M(l) \ RU(-\beta)RH(-\alpha)$. This command set is noted with *m*.

$$\alpha = \begin{cases} \cos^{-1} \frac{x'-x}{\sqrt{(x'-x)^2+(y'-y)^2}} \times 180/\pi \ y' - y \geq 0 \\ -\cos^{-1} \frac{x'-x}{\sqrt{(x'-x)^2+(y'-y)^2}} \times 180/\pi \ y' - y < 0 \end{cases} \alpha \in (-180, \ 180] \tag{1}$$

$$\beta = \cos^{-1} \frac{z'-z}{\sqrt{(x'-x)^2+(y'-y)^2+(z'-z)^2}} \times 180/\pi \ \beta \in [0, \ 180] \tag{2}$$

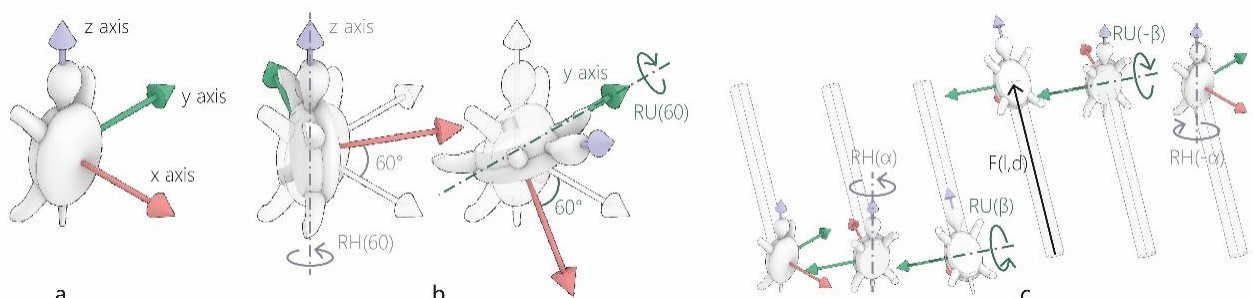

**Figure 5.** Turtle state and Euler rotation in the interpretation of pipelines. (**a**) the initial heading of the turtle in Language XL; (**b**) rotating the turtle's heading along its z and *y* axis with command RH (α) and RU (β), respectively; (**c**) visualization of one set of commands for describing one single pipeline.

The second step is to connect these commands together to describe the topology of the tree. Bracket marks "[" and "]" means push and pop the turtle state in XL language. In other words, the turtle state is temporarily stored at each "[" mark. And the turtle will return to this state when it receives the "]" command. By returning to this state, the temporary storing mark is also cleared. So, what is written inside of a bracket does not affect the geometry by following commands. The overall writing strategy for a tree example is illustrated in Figure 6. Tree sections without sub-branch are simply the conduction of pipeline commands in a growth order: older pipelines before the younger (i.e., see $p_{28}p_{29}p_{30}p_{31}$ in Figure 6). At places where multiple sub-branches occur, except one sub-branch writing in the last, the rest branches should be wrapped in the bracket (see colorful and black brackets in Figure 6). Until now, the topological structure of tree branches has been interpreted from TIM data into Language XL.

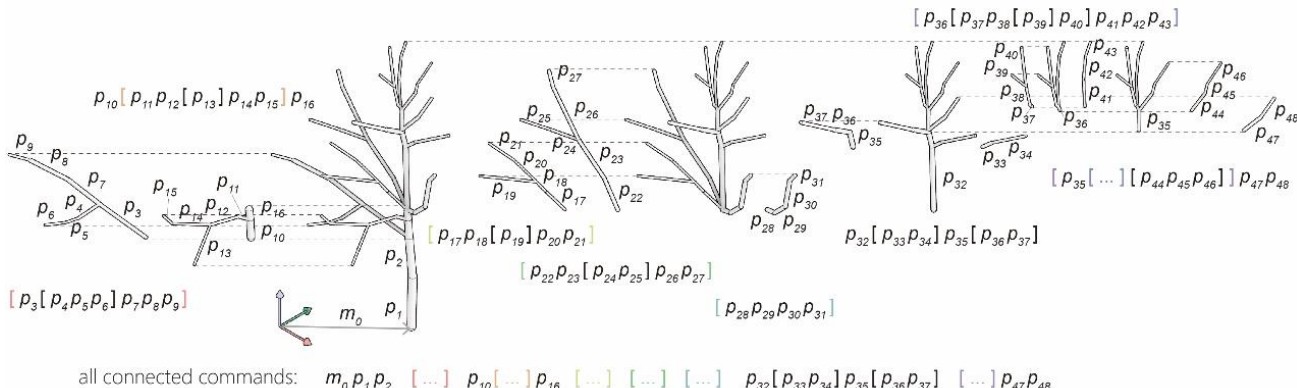

**Figure 6.** Connecting all turtle commands to represent the topology of a tree.

The third step is to enable functions like tree growth in FSPM by adding defined tree organs and parameters to the string. Different simulation purpose requires different definitions and parameters of these organs. Commonly used ones in FSPM are internode, bud, leaf and so on. TIM is a system for broad applications on trees. So, it will not store all detailed information about every tree organ and their possible parameters. TIM should only save the most common information about a tree to enable compatibility between cross-discipline tree applications. Adding unnecessary parameters to TDS means increasing the cost of establishing the database. In consideration of balancing data needs from different professions, internode (trunk and branch segment) and leaf are contained in the first version of TDS (as described in Section 2.1) because they provide a fundamental description of a tree. Their data acquisition methods are also well developed (see Section 2.2.1). Buds, flowers, and fruits, due to their very targeted use, are not documented in the current TDS version. Therefore, such organs must be manually inserted into the interpreted strings for FSPMs. Even though one tree organ-like internode is stored in TDS, some parameters except

its geometry are not compulsory (i.e., the sap flow rate through the internode) in TDS. They can be documented as additional attributes on every geometric element of TIM and then be interpreted into the strings (i.e., "$F(l, d, sapflow\,rate, other\,additional\,attributes\ldots)$"). Some input processes like inserting bud could be, in the future, replaced by programs that make automatic tree organ cognition.

With all three steps, a tree model in TIM is transformed into an FSPM for growth simulation or visualization.

## 3. A Vision of TIM

To enhance the use and applications of TIM in the future, challenges and opportunities are addressed in this section to guide future works.

### 3.1. Development in Data Acquisition and Application

In terms of data acquisition, photogrammetry and TLS were tested by the author to acquire detailed branch geometry (see Section 2.2.1). Both methods can obtain shapes of small branches (with radii smaller than 10 mm) in point clouds. However, a photogrammetry survey has certain requirements of stable soft light, calm air and clear background when taking photos. Without proper training, ordinary users cannot acquire a usable point cloud of trees. Although TLS has a lower skill requirement demand on the user, its high costs limit its share and application scenes in the industry. Moreover, point clouds are redundant for getting only the tree geometry. Storage and transmission of such heavily redundant data have little economic value. Not to mention the massive computing resources used for generating and processing the point cloud. Therefore, an in-time skeletonization solution should be considered, where the point cloud is only a temporary media while only cylindrical pipelines for branch geometries are stored. This solution is possibly combined with LiDAR SLAM. Robotic arms carry a portable LiDAR module going around branches to acquire small sections of their geometries and translate them into pipelines immediately.

To allow wider applications on TIM, data acquisition through other approaches about trees should be able to add to the TIM database. For example, an inclinometer and elastometer were used in pull tests for measuring the deformation of tree trunks under a given force [47,48]. These tests ensure tree stability against windstorms. Data in such tests can be added to branch properties and with these data, structural analysis can be implemented on TIM. In this way, tree failures can be warned ahead of meteorological disasters (in reference to Chan and Eng [153]).

For another application example, the leaf area density of trees in voxels is estimated by their branch pipelines in TIM. Because the shape and density of the tree canopy is very closely related to branch geometry. When this branch-to-leaf relationship for different species can be quantitatively described, by scanning the topological geometry of branches in winter, both branch and canopy data can be estimated. This would spare the work of scanning the canopy again in summer. More importantly, such an application associates the manipulation of branch geometries with targeted functions provided by the canopy.

These suggested applications pave the way for a design workflow for treating trees as a core element in the built environment (see Section 3.3).

### 3.2. Merging the Bottom-Up Simulation with the Global Status of the Tree

Tree growth models in the field of forest science are based on collected data containing a global description of trees under different environmental conditions. So, their models are relatively reliable in forecasting global indicators of trees like biomass and DBH in regard to different ages and species etc. of a tree. This is the advantage of top-down modeling. On the contrary, plant models in the field of FSPM commonly use plant organs (such as internode and bud) as agents to simulate plant behaviors. These are bottom-up models. They have the advantage of reproducing featured patterns in plant growth (see Section 1.2.3). However, their global performances rely on parameters in their physiological process. Some of the settings may not match well with the empirical data. Simulations using TIM in branching

scales are also agent-based (bottom-up) models. So, it is a challenge to match the results of this method to empirical data on a global scale. The gap and difference between these two approaches are expected to be better observed and studied if more tree data are collected and shared under the proposed TDS. These findings can lead to (1) modifying equations for improving the quality of bottom-up models, (2) inspiring new theories to explain and model emergent behaviors on trees, and (3) explaining with a deeper insight into patterns in top-down models. One day, tree growth simulation might merge bottom-up and top-down approaches as a unified system.

### 3.3. Design Workflow Assisted by TIM

A design workflow in a project is as important as the design itself. Achieving multifunctional use of urban trees requires more than datasets and methods. Designers and planners must be able to engage in tree planning by taking advantage of digital tools. As voxels are intuitive in showing spatial distributions of leaves and are used in environmental simulations [26], we hold a strong vision that designers could design tree canopies and their rough density in voxels for urban space (see Figure 7). This design serves as a target parameter for tree status in the long term (i.e., 10–20 years).

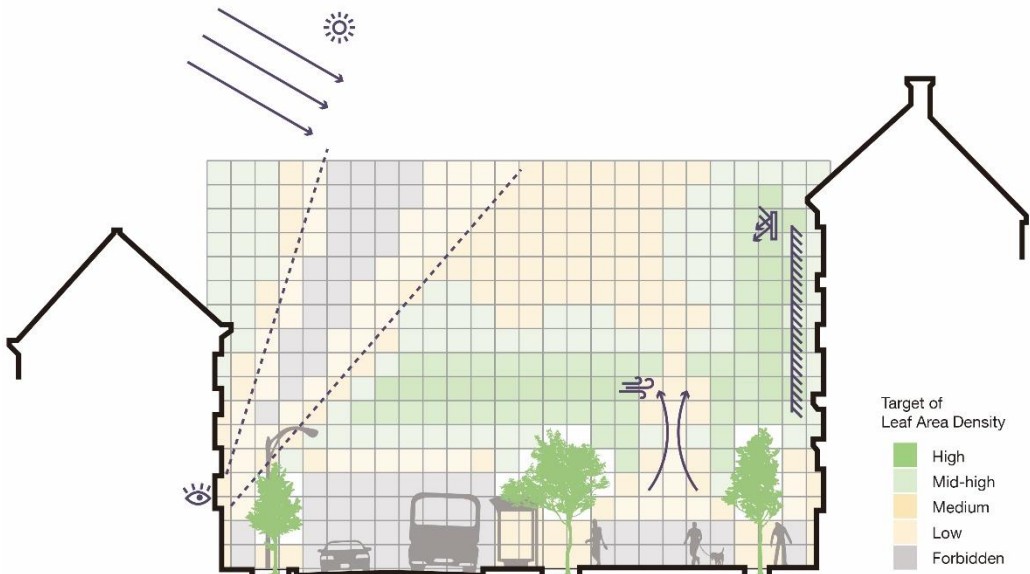

**Figure 7.** An example of designing street green space with voxels containing leaf area density: the north-facing apartments have access to sunlight and sky view; the motorway and south-facing façade are shaded against high radiation; ventilation at the sidewalk is not blocked. Designers can set target values such as rough leaf area densities to these voxels for various design purposes.

To reach such target parameters, trees must grow under specific manipulation and guidance (especially pruning and bending). Each manipulation of trees impacts the later outcome of leaf area density. So, a feedback workflow is required, consisting of scanning, simulation and decision making for maintenance strategy (see Figure 8): (1) based on the current tree status, boundary conditions are input to simulate the future status of this tree; (2) this simulation will be examined and corrected based on the tree growth in reality; (3) this simulation will also be compared to target parameters in design; (4) different manipulation methods will be virtually tested in TIM to get the best solution for approaching the design target. These four steps are repeated multiple times until getting close to the target parameters. The target parameter, boundary conditions as well as simulation method enable modification during any step in the loop. These changes will not affect previously made manipulations on the tree because tree scans and simulations can be performed at any point again to restart the loop. Every step in this workflow relies on data and methods in TIM. Simulating the tree's reaction to manipulations will be the focus of the next step

in this research. A decision-making mechanism to deal with possible conflicts between short-term and long-term outcomes would be a step further. With these steps, the proposed design workflow is developed closer to real application in industry.

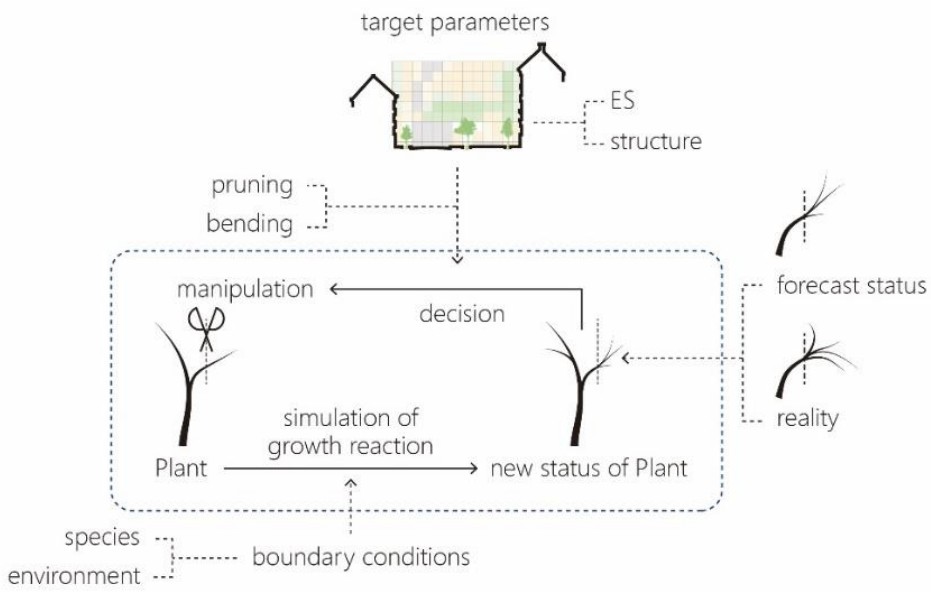

**Figure 8.** A proposed feedback structure about the iterative survey-simulation-manipulation procedure for designing with trees.

## 4. Conclusions

In both architecture design competition and practice, integrating trees in building and engineering systems has become the trend worldwide. Trees differ from traditional building materials by their dynamic growth and requirements for constant maintenance. Therefore, efficiently managing trees in a built environment and enabling their multifunctional uses is the key.

In reference to BIM, we proposed TIM (Tree Information Modeling), serving as a bridge between trees in reality and digital applications in multiple professions, including arboriculture, forestry science, biology, animation and the building industry. A complete TIM model consists of information tags and geometric representations for its root, branch (incl. trunk and aerial root), and canopy respectively. It is described in this paper how these data could be collected with various approaches (such as LiDAR scanning, skeletonization and additional measurement). It is also shown how the topological geometry of branches in the TIM model is interpreted to L-system for implementing widely established tree simulations based on that system.

By bringing all related knowledge and data together, TIM can achieve an accurate evaluation of trees grown in various specially trained forms in the city. For gardening companies, nurseries, and urban planners, they can acquire prediction of water and nutrition consumptions through the lifecycle of the trees, estimating their economic benefits and financial expenses; for constructors and civil engineers, TIM can provide them information regarding a minimum space needed for tree canopy and root growth. For arborists, risks of failure in extreme weather or when trees suffer accident damage can be assessed from TIM data. For studying urban forestry, TIM has accurate geometrical data to estimate trees' impact on microclimate through cooling and evaporation. For architects and landscape designers, TIM can assist with species selection, planting layout and branch configuration. With TIM, trees can interact more with other artificial materials and components without causing unpredictable consequences. All these applications then lead to a longer life expectancy of trees in a densely built urban environment and enable the design and management of gray and green infrastructure in harmony.

Despite these benefits of TIM, limitations were also seen when such standards were applied to industry (in reference to limitations of BIM in implementation [154]). Firstly, it enforces a higher learning and training cost for all participants working with TIM. A tree nursery, for example, may not immediately benefit much from TIM data sources but can invest more to adapt their original data storage form and workflow. Secondly, when tree data are commonly structured and packaged as TIM suggests, it raises a higher threshold (especially for the public) to retrieve, interpret and reuse the data. Lastly, it lacks contractual arrangements yet to specify the property right, legal access and liabilities for tree data.

Finally, efforts still need to be made in efficient data acquisition and discovery of more application scenes on TIM. The gap between the results of branch-scale simulation and a global tree status must be matched. With all these efforts, the goal is to achieve an iterative workflow to manage urban trees towards a design proposal quantified with target parameters.

**Author Contributions:** Conceptualization, Q.S. and F.L.; methodology, Q.S.; software, Q.S.; writing— original draft preparation, Q.S.; writing—review and editing, Q.S., T.R., A.D. and F.L.; visualization, Q.S.; supervision, F.L. and T.R.; project administration, Q.S. and F.L.; funding acquisition, F.L. and T.R. All authors have read and agreed to the published version of the manuscript.

**Funding:** This study is funded by the German Research Foundation (DFG) under the project numbers DFG-GZ: LU2505/2-1 RO4283/2-1 and PR 292/23-1.

**Institutional Review Board Statement:** Not applicable.

**Informed Consent Statement:** Not applicable.

**Data Availability Statement:** Not applicable.

**Acknowledgments:** Our previous work cooperated with Wilfried Middleton about topological skeleton abstraction for inosculated structures inspired the idea of TIM. Haoyu Fang provides the primary data of the tree in Figures 4 and 6.

**Conflicts of Interest:** The authors declare no conflict of interest.

## Abbreviations

| Abbreviations | Full Term |
| --- | --- |
| BIM | Building Information Modeling |
| CAD | Computer Aided Design |
| CIM | City Information modeling |
| DBH | Diameter at Breast Height |
| ESS/ES | Ecosystem Services |
| FSPM | Functional Structural Plant Model |
| GIS | Geographic Information System |
| IoT | Internet of Things |
| LAD | Leaf Area Density |
| LAI | Leaf Area Index |
| LiDAR | Light Detection And Ranging |
| LIM | Landscape Information Modelling |
| LoD | Level of Detail |
| QSM | Quantitative Structure Models |
| SfM | Structure from Motion |
| SLAM | Simultaneous Localization and Mapping |
| TDS | Tree Description System |
| TIM | Tree Information Modeling |
| TLS | Terrestrial Laser Scanning |
| UHI | Urban Heat Islands |

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
