# Peer review of "Tree Information Modeling: A Data Exchange Platform for Tree Design and Management"

_forests, doi:10.3390/f13111955_

Round 1

Reviewer 1 Report

Dear authors,

Thank you for sending this paper to the journal Forests. The subject of the study is very interesting and the implementation of a complete protocol that couples tree design and management to study the subject of the tree biomimetic modeling is of great interest. 

Comment 1: The title, abstract correspond to the aims and objectives of the manuscript. The opponent thinks that it would be appropriate to remove 'Digital Green Technology', and add to the keywords: green infrastructure.

Comment 2: '1.1. aim of this study' part include too many paragraphs. Perhaps more level 3 title like '1.1.1 ...' .

Comment 3: More scenes or context should be mentioned, such as tree in national park, roof greening, scenic area, etc. Perhaps some recent refs are, for example: 1) Impacts of national park tourism sites: a perceptual analysis from residents of three spatial levels of local communities in Banff national park. Environment Development and Sustainability 2022 24(4), 3126-3145; 2) Impact of Different Combinations of Green Infrastructure Elements on Traffic-Related Pollutant Concentrations in Urban Areas. Forests 2022, 13(8), 1195; 3) Before Becoming a World Heritage: Spatiotemporal Dynamics and Spatial Dependency of the Soundscapes in Kulangsu Scenic Area, China. Forests 2022 13(9):1526.

Comment 4: You mentioned Urban forestry many times. But there are missing several publications from Konijnendijk (only one in this version). Furthermore, For the topic of urban forestry, i wonder mention publications from the journal Forests. Perhaps some recent refs are, for example: 1) Assessment of Above-Ground Carbon Storage by Urban Trees Using LiDAR Data: The Case of a University Campus. Forests 2021, 12(1), 62. 2) Perceived loudness sensitivity influenced by brightness in urban forests: a comparison when eyes were opened and closed. Forests, 11(12):1242; 3) National Forest Parks in China: Origin, Evolution, and Sustainable Development. Forests 2019, 10(4), 323.

Comment 5: Figure 8. Did different densities affect your results (decisions)?

Comment 6: You mentioned limitation many times. However, I still consider you need to summarize the technical limitations again (especially for the application scenario) before conclusion part.

Comment 7: The format of references does not seem to match the requirements of the journal Forests.

Comment 8: minor things. You used too long sentences in some parts. After reaching the end of the sentences, the reader cannot remember how the sentence started.

Best regards

Author Response

Dear reviewer,

thank you for your suggestions and feedbacks. Please find our detailed replies to each comment below:

Comment 1: We agree. The key word “digital green technology” is now replaced with “urban green infrastructure”.

Comment 2: Thanks for the suggestion. We have added 5 subtitles to section 1.1.

Comment 3 and 4: Thanks for the helpful references. Especially the papers by Konijnendijk. Papers regarding the carbon storage assessment, soundscape, and traffic related pollutant are also of great values to this paper. We have included them now in the citations. For the papers regarding national parks and scenic areas, we decided not to cite them in this paper, because we are more focused on the problems in densely built urban areas. Citing studies about natural reservations in less built areas might distract our study aim.

Comment 5: Sorry, we do not fully understand this question. To clarify a possible misunderstanding, we did not specify how the decisions regarding leaf density should be made. We only suggested in figure 8 that it is possible to design future urban green infrastructure with voxels showing different densities. We rephrase the figure capital a bit to make it clearer.

Comment 6: The TIM concept itself is a theoretical framework. It is, therefore, not limited by any technical limitations. The limitations in its industrial implementations, following your advice, are added in the conclusion (the second last paragraph).

Comment 7: The reference format is revised.

Comment 8: A few long sentences are divided into shorter sentences.

Wish above replies could answer all your questions and advices.

Best regards,

Qiguan Shu on behalf of all the authors

Reviewer 2 Report

The comments are in the document attached.

Author Response

Dear reviewer,

thank you for your suggestions and feedbacks. Please find our detailed replies to each comment below:

page 1

Comment 1: We believe it is a trend worldwide. “Worldwide” is added to the sentence.

Comment 2: “Poor” is rephrased with “limited by unmatched scales and formats”.

Comment 3: We would like to keep “Tree Information Modeling” as one of the keywords because it is a novel phrase by us. We wish to ensure this phrase easily retrieved with this paper.

Comment 4 and 5: Reference format has been revised through the paper.

Page 2

Comment 1: The long caption of figure 1, 5 and 9 has been moved to the main text.

Comment 2: “RENAZCA” is not an abbreviation, but the name of the project used by the landlord.

Page 3

Comment 1: Thanks for point it out. “25m” -> “25 m”.

Page 4

Comment 1: This is our own Statement. Sorry if this point sounds too strong and subjective. We rephrase “Poor” with “limited owing to unmatched scales and formats” now.

Comment 2: We added “(see dark orange rectangles)” to guide the readers finding the “multiple decisions” in the figure.

Page 5

Comment 1: All chapter titles are starting with capital letters now.

Page 6

Comment 1: “UDFA” -> “USDA”

Comment 2: Thanks for the advice. “carbohydrates” -> “nutrients”

Page 12

Comment 1: We had the list for all abbreviations used in the paper at the end. So, we add in the table caption “(abbreviations listed at the end of the article) ”.

Page 19

Comment 1: Yes, we agree. In the text, “periodic maintenance” can be very frequent. “periodic” -> “constant”.

Page 20

Comment 1: Sorry for missing out this part. I was not aware that this information was not automatically transferred from the submission system to the template. They have been added to the newest word file by now.

Comment 2: Citations have been changed to format used by Forests Journal.

Wish above replies could answer all your questions and advices.

Best regards,

Qiguan Shu on behalf of all the authors

Reviewer 3 Report

Dear Authors,

After going through your article I have to conclude that you should re-write it in a way that we can see what are really your results.

My advise would be to follow the standard article structure, it means introduction, methodology, results, discussion, conclusions. I can see a lot of literature research and comparison of measurement models but it is didifficult for me to be aware of the fact what are actually your results. This should be also clearly articulated in abstract.

The main weakness of this article is absence of the aplication of the proposed tree model at any study area. In present form, it is very theoretical.

Specific comments:

1. many typological errors, for example lines 32, 178, 179, 265, etc..(Title of the chapter and subchapaters should start with a capital letter)

2. Format of this article does not follow prescribed MDPI Forest format (citations in brackets). Furthermore

3."Conflict of interest" part is not finished and other last standard parts are missing. It seems that this article is somehow not finished

4. Sources of figures are missing. If it is your original figure you should indicate it.

Author Response

Dear reviewer,

thank you for your suggestions and feedbacks.

Regarding the structure of this paper, we sincerely appreciate your advice to follow a normal structure. We would like to share one reason, why we did not structure the paper this way. This paper can be categorized as “manuscripts communicating to a broader audience with regard to research projects financed with public funds” and “manuscripts regarding research proposals and research ideas” (see Aim & Scope page of Forests journal). Therefore, we adopt a structure that may better explains our concept and research scope.

Furthermore, we would like to ask the reviewer to consider that the section 2 is the intellectual “results” at current stage while the section 3 is a “discussion” about potential results in the later phase of this research. This resulted in our decision not to follow the usual structure of a scientific publication.

In response to your concern that we may not have emphasized the benefits and applications of TIM concept enough, we rephrased the third paragraph of the conclusion. We highlight now possible applications of TIM that benefit different fields.

Please find our detailed replies to other comments below:

Comment 1: Chapter titles are all starting with capital letters now.

Comment 2: Citations have been changed to the format used by Forests Journal.

Comment 3: Sorry for this issue. I was not aware that this information was not automatically transferred from the submission system to the template. They have been added to the newest word file by now.

Comment 4: Thanks for this comment. Only figure 1 in our paper contains 4 sub-images from the Internet under the Creative Commons licenses. The notes are added to the figure caption. Others are own-made images. Referring to some newest articles in Forests Journal, they do not need a further indication like “own image”.

Wish above replies could answer all your questions and advices.

Best regards,

Qiguan Shu on behalf of all the authors

Round 2

Reviewer 3 Report

The manuscript has been improved